# Integrated Proteomics and Metabolomics Analysis Provides Insights into Ganoderic Acid Biosynthesis in Response to Methyl Jasmonate in *Ganoderma Lucidum*

**DOI:** 10.3390/ijms20246116

**Published:** 2019-12-04

**Authors:** Ai-Liang Jiang, Yong-Nan Liu, Rui Liu, Ang Ren, Hong-Yu Ma, Lie-Bo Shu, Liang Shi, Jing Zhu, Ming-Wen Zhao

**Affiliations:** 1Key Laboratory of Agricultural Environmental Microbiology, Ministry of Agriculture, Microbiology Department, College of Life Sciences, Nanjing Agricultural University, Nanjing 210095, China; aljiang@njau.edu.cn (A.-L.J.); ruiliu@njau.edu.cn (R.L.); angren@njau.edu.cn (A.R.); shiliang@njau.edu.cn (L.S.); jingzhu@njau.edu.cn (J.Z.); 2International Cooperation Base of Science and Technology Innovation on Forest Resource Biotechnology of Hunan Province, Central South University of Forestry & Technology, Changsha 410004, China; ynliu@csuft.edu.cn; 3College of Plant Protection, Nanjing Agricultural University, Nanjing 210095, China; mahongyu@njau.edu.cn; 4Shanghai Luming biotechnology co., ltd, Shanghai 201114, China; shulb@lumingbio.com

**Keywords:** iTRAQ, metabonomics, ganoderic acid, methyl jasmonate, metabolic rearrangement

## Abstract

*Ganoderma lucidum* is widely recognized as a medicinal basidiomycete. It was previously reported that the plant hormone methyl jasmonate (MeJA) could induce the biosynthesis of ganoderic acids (GAs), which are the main active ingredients of *G. lucidum*. However, the regulatory mechanism is still unclear. In this study, integrated proteomics and metabolomics were employed on *G. lucidum* to globally identify differences in proteins and metabolites under MeJA treatment for 15 min (M15) and 24 h (M24). Our study successfully identified 209 differential abundance proteins (DAPs) in M15 and 202 DAPs in M24. We also identified 154 metabolites by GC–MS and 70 metabolites by LC–MS in M24 that are involved in several metabolic pathways. With an in-depth analysis, we found some DAPs and metabolites that are involved in the oxidoreduction process, secondary metabolism, energy metabolism, transcriptional and translational regulation, and protein synthesis. In particular, our results reveal that MeJA treatment leads to metabolic rearrangement that inhibited the normal glucose metabolism, energy supply, and protein synthesis of cells but promoted secondary metabolites, including GAs. In conclusion, our proteomics and metabolomics data further confirm the promoting effect of MeJA on the biosynthesis of GAs in *G. lucidum* and will provide a valuable resource for further investigation of the molecular mechanisms of MeJA signal response and GA biosynthesis in *G. lucidum* and other related species.

## 1. Introduction

*Ganoderma lucidum* is a fungus historically used in traditional herb medicines and supplements in China and other parts of East Asia [1,2]. The major secondary metabolites of *G. lucidum* are lanostane triterpenoid natural products, and more than 150 lanostanes, including ganoderic acids (GAs), which are the main active ingredients, have been isolated from *Ganoderma* spp. GAs. The main active ingredients of *G. lucidum* exhibit various biological activities, such as anti-oxidation, anti-virus and anti-tumor activities [3]. The content of GAs is also an important quality index of *G. lucidum* [4,5]. However, the content of GAs is very low, resulting in high pricing and limited use of GA products. To date, methods to promote GA biosynthesis have been studied, and it has been proven that different carbon sources, nitrogen sources and initial pH have significant effects on GA biosynthesis [6,7]. The addition of some small molecular compounds to mycelium, such as phenobarbital [8], acetic acid [9], and salicylic acid [10], can also significantly improve the content of GAs in *G. lucidum*. Further studies have found that some signaling molecules, such as calcium, nitric oxide, reactive oxygen species (ROS), phospholipids, hydrogen sulfide are involved in the regulation of environment-induced GA biosynthesis [10,11,12,13,14,15]. However, the regulatory mechanism of GA biosynthesis has not been determined to date because the environmental regulation mechanism of microbial secondary metabolism is often a complex regulatory network. Therefore, improving the yield and clarifying the biosynthesis mechanism of GAs are critical to further enhance the medicinal value and commercial value of *G. lucidum*.

Jasmonic acid (JA) and its derivative, methyl jasmonate (MeJA), are hormonal cues released by plants that signal defense responses to curb damage from biotic and abiotic stresses [16]. In plants, JA and its derivatives are involved in the regulation of many developmental processes, including male fertility [17], fruit ripening [18], and root growth [19]. In animals, MeJA has shown growth inhibition and anti-cancer effects on a variety of cancer cells, such as leukemic cells, cervical cancer cells, lymphocytic leukemia cells, in vitro and in vivo [20,21,22]. Compared with plants and animals, there is a little research on MeJA in fungi. However, it has also been found that MeJA directly inhibits the mycelial spread of postharvest pathogens and increases the population of *Cryptococcus laurentii* [23] and stimulates secondary metabolites aflatoxin B1 biosynthesis by *Aspergillus parasiticus* [24]. Our previous research indicated that treatment of *G. lucidum* with MeJA could increase the content of GAs by 28.6% compared with untreated samples [25]. Further research based on a cDNA-amplified fragment length polymorphism (cDNA-AFLP) analysis revealed that 390 transcriptionally derived fragments (TDFs), including GA synthesis genes, ROS scavengers, and Ca^2+^ sensor homologues, were involved in the regulatory network of GA biosynthesis [26]. Recently, we demonstrated that an NADPH oxidase-dependent ROS burst is important for the regulation of GA biosynthesis elicited by MeJA treatment [27]. All these studies discovered new candidate genes involved in the regulation of GA biosynthesis in *G. lucidum*. Revealing the regulatory mechanism of MeJA on GA biosynthesis is of great significance for comprehensively analyzing the environmental regulatory network of GA biosynthesis.

Omics-profiling techniques have proven to be powerful tools for unravelling complex biological processes and have been successfully utilized in fungi. Comparative proteomics based on 2-DE gels was conducted to investigate the change in protein expression of two *Ganoderma* species while interacting with oil palm root in vitro [28]. Entire secretomes of *G. lucidum* cultivated in sugarcane bagasse were identified by liquid chromatography–tandem mass spectrometry (LC–MS) [29]. A metabolomics study discovered that 62 features were either newly synthesized or highly produced in the co-culture of *Trametes versicolor* and *G. applanatum* compared with individual cultures [30]. The molecular mechanisms revealed by using a single omics technique are very limited. The integration of omics technologies has the potential to provide a considerably more detailed view of cellular homeostasis than when the technologies are used individually.

In the present study, integrated proteomics and metabolomics was employed on *G. lucidum* to globally identify differences in proteins and metabolites under MeJA treatment in combination with the analysis of protein function and metabolic pathways. This study may help to elucidate the underlying molecular mechanism of MeJA-induced GA biosynthesis in *G. lucidum* and expand our knowledge of the metabolic processes by which filamentous fungi respond to MeJA.

## 2. Results

### 2.1. Screening the Appropriate MeJA Processing Time Points

Previous studies have shown that MeJA has an upregulation effect on many genes related to GA biosynthesis. The change of signal transduction genes (such as *slt2*, *fus*, *pkc*, *ras* and *nox*) to MeJA is one of the most important early events in the response to MeJA treatment [26,27]. Therefore, we selected five signal transduction genes to monitor the early events of the *G. lucidum* response to MeJA. The genes are the following: *slt2*, involved in the regulation of fungal growth, cell wall integrity, oxidative stress, and ganoderic acid biosynthesis in *G. lucidum* [31]; *fus*, involved in the response to pheromone stimulation [32]; *pkc*, a component of the DAG/PKC signal transduction pathway and an important element in the regulation of intracellular signaling networks [33]; *ras*, involved in the early response to extracellular signaling and multi-stress tolerance [34]; and *noxa*, a subunit of membrane-bound fungal NADPH oxidases (Nox) that is involved in the regulation of ROS synthesis and defensive reactions [35]. In early events response to MeJA, signal transduction genes were usually upregulated within an hour [18]. Therefore, we detected the expression changes of those signal transduction genes within 0–240 min. The qTR-PCR results show that compared to the normal growth conditions, quick responses were observed in the five signal transduction genes after MeJA treatment for 15 min (Figure 1A). MeJA led to a significant increase (*p* < 0.01) by approximately 3.1-, 2.1-, 3.9-, and 5.8-fold at 15 min in the *slt2*, *fus*, *noxa*, and *ras* gene expression levels, respectively. These transcriptional fluctuations quickly return to normal levels. However, there was no significant change in the transcription level of *pkc* (Figure 1A). These results indicate that MeJA treatment for 15 min (M15) rapidly induced the expression of several kinases and regulatory genes.

To investigate the potential role of MeJA in GA biosynthesis, we also investigated the expression levels of four key genes in GA biosynthesis, *hmgr* (encoding 3-hydroxy-3-methylglutaryl coenzyme A reductase), *sqs* (encoding squalene synthase), *osc* (encoding lanosterol synthase), and *fps* (encoding farnesyl pyrophosphate synthase), by qRT-PCR. Previous studies have shown that *hmgr*, *sqs*, and *osc* genes in GA biosynthesis were upregulated at 24–48 h under different conditions [8,12,26,27]. Therefore, we detected the expression changes of genes related to GA biosynthesis within 0–48 h. Treatment with MeJA for 24 h led to a significant increase (*p* < 0.01) by approximately 3.5-, 2.0-, and 1.5-fold in the *hmgr*, *sqs*, and *osc* gene expression levels, respectively, compared with the non-MeJA treatment (Figure 1B). There was no significant change in the transcription level of *fps*. These results are largely consistent with the findings of previous studies [26] and indicate that MeJA treatment for 24 h (M24) is the best condition to induce GA biosynthesis.

### 2.2. Overview of Quantitative Proteomics Analysis

Proteins were extracted from control and treated (M15 and M24) samples and processed with iTRAQ technology. A total of 307,493 spectra were identified by an iTRAQ-LC-MS/MS proteomic analysis in the present study. After the data were filtered, a total of 106,017 unique spectra and 26,337 unique peptides were obtained. To avoid identification errors, 5059 proteins with FDR (false discovery rate) ≤0.01 were further confirmed using the *G. lucidum* Genome Annotation Project database (http://www.herbalgenomics.org/galu). According to unused ≥1.3 and peptides (95%) ≥2, a total of 3918 proteins were identified for further comparative analysis. Among the statistically significant proteins detected by the ANOVA test (*p* < 0.05), protein abundances that changed less than 1.5-fold and *p* > 0.05 were discarded. Following this criterion, we detected a total of 352 proteins that were differentially abundant in M15 and M24. In addition, the Venn analysis results showed that there were 59 differential abundance proteins (DAPs) of M15 and M24 in common (Appendix A).

### 2.3. Characteristics of DAPs in M15

A total of 209 proteins are differentially abundant in M15. Of these proteins, 118 (Appendix A) were upregulated and 91 proteins (Appendix A) were downregulated in response to MeJA treatment for 15 min. The 209 DAPs of M15 were analyzed using a bioinformatics approaches to extract information related to the involved functions by GO. The enrichment analysis showed that the 209 DAPs were significantly enriched (*p* < 0.05) in 366 biological processes, 46 cell components, and 140 molecular functions (Appendix A). In biological processes, small molecular metabolic processes were the most representative term, followed by organonitrogen compound metabolic processes and single-organism catabolic processes (Appendix A). In cell components, the most representative term was followed by cytosol and ribosome (Appendix A). In molecular functions, catalytic activity was the most representative term followed by heterocyclic compound binding and oxidoreductase activity (Appendix A). The KEGG analysis revealed that the 209 DAPs were enriched in 16 pathways (*p* < 0.01) (Appendix A). The representative terms included metabolic pathways, biosynthesis of secondary metabolites, carbon metabolism, fructose and mannose metabolism, and pyruvate metabolism (Appendix A). The detailed data of 209 DAPs are shown in Appendix A. 

Further analysis of the upregulated DAPs revealed that proteins related to the secondary metabolic synthesis pathway of terpenes were significantly upregulated. Cytochrome P450 61 (GL27042-R1_1, 3.233), sterol 24-c-methy-ltransferase (GL28304-R1_1, 1.808), phosphoevalonate kinase (GL17808-R1_1, 3.306), lanosterol 14-alpha demethylase (GL28943-R1_1, 5.514), farnesyl pyrophosphate synthase (GL22068-R1_1, 1.787), and sterol-4-alpha-carboxylate 3-dehydrogenase (GL16838-R1_1, 2.006) were all increased by different degrees (Appendix A). These results suggest that MeJA treatment has a significant effect on the triterpene synthesis pathway in a short time, which also explains why the multiple related secondary metabolites in the M24 were increased (as described in the metabolic changes section).

Cytochrome c oxidase (GL23456-R1_1, 2.856) and peroxiredoxin (GL21226-R1_1, 4.588) which play roles in cell protection against oxidative stress and as sensors of hydrogen peroxide-mediated signaling events were significantly upregulated (Appendix A). Catalase T (GL22189-R1_1, 0.223), the predominant scavenger of hydrogen peroxide, was quickly consumed (Appendix A). These results indicate that the antioxidant reductase system was activated and hint that the ROS signal was activated in the early response of MeJA processing (M15). On the other hand, the functional proteins related to glycolysis, gluconeogenesis, and TCA cycle were upregulated (Appendix A), including isocitrate lyase (GL21539-R1_1, 10.608), hexokinase-2 (GL20491-R1_1, 1.683), D-xylulose reductase (GL22360-R1_1, 5.31), phosphoglucomutase 2 (GL24280-R1_1, 2.136), pyruvate carboxylase 1 (GL21673-R1_1, 1.874), phosphoenolpyruvate carboxy kinase (GL25620-R1_1, 5.103), malate synthase 2 (GL23465-R1_1, 3.79), and NAD-dependent malic enzyme (GL22666-R1_1, 2.504); all these results indicate an accelerated rate of energy metabolism in M15.

Signal perception and transduction pathways were quickly initiated. As shown in Appendix A, calmodulin (GL23980-R1_1, 1.808) which mediates the control of a large number of enzymes, ion channels, and other proteins by Ca^2+^, was upregulated. The alcohol dehydrogenase family were generally upregulated, including alcohol dehydrogenase 1 (GL24208-R1_1, 3.082), alcohol dehydrogenase 2 (GL20764-R1_1, 7.821), alcohol dehydrogenase 3 (GL22020-R1_1, 4.721), and alcohol dehydrogenase 5 (GL24163-R1_1, 2.188). The upregulation of these stress response proteins suggests that the appearance of MeJA may be an abiotic stress signal for *G. lucidum* hyphae.

On the contrary, a large number of ribosome related proteins were significantly downregulated. As shown in Appendix A and Appendix A, nine 60S ribosomal proteins, and 18 40S ribosomal proteins were significantly reduced. The most severely reduced ribosomal proteins were 40S ribosomal protein S7-B (GL31438-R1_1, 0.221) and 60S ribosomal protein L9-A (GL31416-R1_1, 0.271). These reductions would adversely affect 60S and 40S subunits’ biogenesis, and inevitably lead to the stagnation of protein biosynthesis. In addition, DNA replication, transcription, translation, and cell division related proteins were inhibited. In detail (Appendix A), DNA replication factor C subunit 1 (GL20422-R1_1) was downregulated 0.525-fold; transcription regulatory protein SIN3 (GL24127-R1_1) was reduced by 0.524-fold; pre-mRNA-splicing factor PRP46 (GL25322-R1_1) was downregulated by 0.489-fold; cold sensitive U2 snRNA suppressor 1 (GL24643-R1_1), which is essential for spliceosome assembly, was downregulated by 0.265-fold; RNA annealing protein YRA1 (GL22546-R1_1) was downregulated by 0.578-fold; mRNA stability protein IGO2 (GL19525-R1_1) was downregulated by 0.383-fold. In addition, eIF-2-alpha kinase activator GCN1(GL24443-R1_1) was downregulated by 0.441-fold, and protein GCN20, which is involved in the regulation of protein synthesis, was downregulated by 0.496-fold. On the other hand, cell division control protein 3, (GL30629-R1_1) decreased by 0.372-fold and cell division control protein 10 (GL24518-R1_1) decreased by 0.518-fold, which suggests that the cell cycle might be delayed by the MeJA (Appendix A).

### 2.4. Characteristics of DAPs in M24

A total of 202 proteins were differentially expressed in response to MeJA treatment for 24 h. A total of 54 DAPs (Appendix A) were upregulated and 148 DAPs (Appendix A) were downregulated. The enrichment analysis showed that the 202 DAPs were significantly enriched in 337 biological processes, 42 cell components, and 161 molecular functions (Figure 2). In biological processes, organonitrogen compound metabolic processes were the most representative term, followed by small molecular metabolic processes and catabolic processes (Figure 2, left). In cell components, intracellular was the most representative term followed by cytoplasm and ribonucleoprotein granule (Figure 2, center). In molecular function, catalytic activity was the most representative term followed by ion binding and oxidoreductase activity (Figure 2, right). The KEGG analysis revealed that the 202 DAPs were enriched in 21 pathways (Figure 2 and Figure 3A). The most significant (*p* < 0.01) 10 pathways were biosynthesis of secondary metabolites followed by biosynthesis of antibiotics, metabolic pathways, glycolysis/gluconeogenesis, fatty acid degradation, butanoate metabolism, pyruvate metabolism, fructose and mannose metabolism, tryptophan metabolism and biosynthesis of amino acids (Figure 3B). The detailed data of 202 DAPs are shown in Appendix A.

Further analysis of 54 proteins in the upregulated group revealed that the secondary metabolic synthesis pathway of terpenes remains upregulated. In detail, cytochrome P450 61 (GL27042-R1_1, 5.427) involved in steroid biosynthesis, hydroxymethylglutaryl-CoA synthase (GL24922-R1_1, 1.694) involved in butanoate metabolism, farnesyl pyrophosphate synthase (GL22068-R1_1, 7.275) involved in terpenoid backbone biosynthesis, alpha-factor-transporting ATPase STE6 (GL23892-R1_1, 3.638) involved in the transportation of the farnesyl-derivation of the A-factor pheromone were all increased (Appendix A). The promotion effect of MeJA on triterpenoids of GAs was confirmed again, and these results also conformed to the increased GAs content phenotype.

The cell wall structure and morphological maintenance related proteins were upregulated in M24. In detail (Appendix A), 1,3-beta-glucanosyltransferase GAS1 (GL29873-R1_1) participates in the cell wall biosynthesis and morphogenesis was increased by 2.374-fold. Cytochrome P450-DIT2 (GL15986-R1_1) involved in spore wall maturation was increased 2.42-fold. Killer toxin-resistance protein KRE5 (GL22536-R1_1) participates in fungal cell wall (1, 6)-beta-D-glucan synthesis and cell development was increased by 2.167-fold. All these results suggest that the cell wall structure and function of *G. lucidum* mycelia might be disturbed after MeJA treatment for 24 h. On the other hand, endoplasmic reticulum mannosyl-oligosaccharide 1,2-alpha-mannosidase (GL30115-R1_1, 3.987) involved in the degradation of misfolded glycoproteins and plays an important role in protein glycosylation and quality control was significantly upregulated. Thiol-specific monooxygenase (GL15434-R1_1, 4.32) required for the correct folding of disulfide-bonded proteins was significantly increased. These results indicate that protein quality control was strengthened after MeJA treatment for 24 h.

However, a bioinformatics analysis of 148 proteins in the downregulated group of M24 showed that many functional proteins are involved in glycolysis, gluconeogenesis, and hexose metabolism (Appendix A), which was totally different from the result of M15 in which an accelerated rate of energy metabolism was observed. The related downregulated DAPs included sorbitol dehydrogenase 1 (GL17001-R1_1, 0.448), hexokinase-2 (GL20491-R1_1, 0.421), fructose-bisphosphate aldolase (GL25136-R1_1, 0.55), glycogen phosphorylase (GL21375-R1_1, 0.572), glycerol 2-dehydrogenase (GL31363-R1_1, 0.514), enolase 1 (GL30114-R1_1, 0.411), glycerol-3-phosphate dehydrogenase (GL29426-R1_1, 0.596), triosephosphate isomerase (GL22268-R1_1, 0.514), fatty aldehyde dehydrogenase (GL20474-R1_1, 0.386), D-lactate dehydrogenase 1 (GL26490-R1_1, 0.232), and pyruvate decarboxylase isozyme 1 (GL23873-R1_1, 0.389). These results suggest that MeJA treatment for 24 h might have a negative effect on glucose metabolism and the cellular energy supply level would be depressed.

On the other hand (Appendix A), 2-isopropylmalate synthase LEU4 (GL19766-R1_1) involved in leucine synthesis pathway was decreased by 0.469-fold. Homocysteine methyltransferase (GL31662-R1_1) involved in methionine synthesis pathway was decreased by 0.34-fold. Dihydroxy-acid dehydratase ILV3 (GL19372-R1_1), involved in the valine synthesis pathway, was decreased by 0.544-fold. L-2-aminoadipate reductase (GL24538-R1_1, 0.388) and saccharopine dehydrogenase (GL22111-R1_1, 0.327), involved in lysine synthesis pathway, were both downregulated. In addition to the above, acetylornithine aminotransferase ARG8 (GL19179-R1_1, 0.44) and argininosuccinate synthase ARG1 (GL29631-R1_1, 0.519) involved in arginine anabolism, histidine biosynthesis trifunctional protein HIS4 (GL31286-R1_1, 0.557) involved in L-histidine synthesis, and multifunctional tryptophan biosynthesis protein TRP3 (GL20235-R1_1, 0.318) involved in tryptophan synthesis were all downregulated. All these results indicate that amino acid biosynthesis was decreased after MeJA treatment for 24 h.

Even worse, some amino acid tRNA ligases associated with amino acid transport and protein synthesis were also downregulated (Appendix A), which included isoleucine-tRNA ligase (GL25741-R1_1, 0.425), serine-tRNA ligase (GL15281-R1_1, 0.626), threonine-tRNA ligase(GL21206-R1_1, 0.511), tyrosine-tRNA ligase (GL27820-R1_1, 0.45), glutamate-tRNA ligase (GL24500-R1_1, 0.517), histidine-tRNA ligase (GL30373-R1_1, 0.537), tRNA-aminoacylation cofactor (GL23744-R1_1, 0.509), dicarboxylic amino acid permease (GL22889-R1_1, 0.459). These results further suggest that a global repression of protein synthesis occurred under MeJA treatment for 24 h.

Ribosome constituent related proteins remained decreased (Appendix A), which included 40S ribosomal protein S2-B (GL19949-R1_1, 0.419), 40S ribosomal protein S17-A (GL30395-R1_1, 0.529), 40S ribosomal protein S18-B (GL25744-R1_1, 0.358), 40S ribosomal protein S19-B (GL28146-R1_1, 0.534), 40S ribosomal protein S6-A (GL22318-R2_1, 0.449), 60S ribosomal protein L5 (GL31426-R1_1, 0.461), 60S ribosomal protein L6-A (GL31415-R1_1, 0.312). These results are consistent with the results of M15 and indicating that MeJA treatment was a negative regulator of ribosomal subunit proteins. In addition, proteins involved in DNA replication, transcription, splicing, translation initiation, and elongation also continued to decline (Appendix A), which included DNA topoisomerase 1 (GL25411-R1_1, 0.349), DNA helicase NAM7 (GL28710-R1_1, 0.426), ATP-dependent RNA helicase FAL1 (GL24146-R1_1, 0.583), the transcription elongation factor 1 subunit CAF16 (GL31272-R1_1, 0.305), transcription elongation factor 1 sub CAM1 (GL23700-R1_1, 0.324), and eIF-2-alpha kinase activator GCN1 (GL24443-R1_1, 0.515). Moreover, pre-mRNA-processing protein PRP40 (GL21804-R1_1, 0.42), pre-mRNA-splicing factor PRP46 (GL25322-R1_1, 0.563) and pre-mRNA-splicing factor SNU114 (GL30704-R1_1, 0.498) were downregulated. The eukaryotic translation initiation factor 3 subunit I (GL23794-R1_1, 0.501), eukaryotic translation initiation factor 3 subunit A (GL23081-R1_1, 0.585), eukaryotic translation initiation factor 3 subunit B PRT1 (GL22820-R1_1, 0.474), and eukaryotic peptide chain release factor subunit 1 SUP45 (GL24973-R1_1, 0.366) were all significantly decreased. All these results suggest that the *G. lucidum* mycelium was in an inhibitive basic physiological state after MeJA treatment for 24 h.

Multiple functional proteins involved in morphological development and cytokinesis were also downregulated (Appendix A). Serine/threonine-protein phosphatase (GL25257-R1_1), involved in the control of glycogen metabolism, meiosis, translation, chromosome segregation, cell polarity, and G2/M cell cycle progression, was decreased by 0.527-fold. Morphogenesis-related protein MSB1 (GL21274-R1_1), involved in polarity establishment, was decreased by 0.549-fold. Serine/threonine-protein kinase CLA4 (GL18548-R1_1) involved in budding and cytokinesis, was decreased by 0.437-fold. Cell division control protein CDC48 (GL21736-R1_1), involved in spindle disassembly at the end of the mitosis and endoplasmic reticulum-associated degradation (ERAD) pathway, was decreased by 0.534-fold. These results reveal that cellular morphological development and polar growth might be disturbed under M24. Combined with the aforementioned downregulation DAPs in M14 and M24, we speculated that MeJA may have a similar effect of “hitting the brakes” on the normal growth of *G. lucidum* hyphae cells.

### 2.5. Characteristics of DAPs in M15 and M24

The Venn analysis results show that there were 59 DAPs of M15 and M24 in common (Appendix A). In terms of triterpene synthesis, cytochrome P450 (GL27042-R1_1, ERG5) and farnesyl pyrophosphate synthase (GL22068-R1_1, ERG 20) were detected and upregulated both in M15 and M24 (Appendix A); this is consistent with the phenotypic results from the measured GAs contents before. Interestingly, hexokinase HXK2 (GL20491-R1_1), glycerol 2-phosphate dehydrogenase GCY1 (GL24473-R1_1), and phosphopropiose isomerase TPI1 (GL22268-R1_1), which are involved in the energy metabolism pathways, were upregulated in M15 but downregulated in M24 (Appendix A). These results suggest that the intracellular energy metabolism pathways might be rearranged from basic metabolism to secondary metabolism in the MeJA induction proceeding (Figure 6 and Appendix A).

On the other hand (Appendix A and Appendix A), phosphatidylserine decarboxylase PSD2 (GL15153-R1_1) was upregulated both in M15 and M24, suggesting that MeJA may also affect the glycerophospholipid homeostasis. Iron transport multicopper oxidase FET3 (GL16398-R1_1) involved in iron ion transmembrane transport, had always been upregulated. However, catalase T CTT1 (GL22189-R1_1) was always downregulated, suggesting that catalase T may play an important role in dealing with ROS caused by MeJA both in M15 and M24. Notably, ribosomal proteins tended to be downregulated both in M15 and M24, such as ribosome 40S small subunit protein RPS17A, RPS18B, RPS19B, RPS6A, and 60S large subunit protein RPL6A, which indicates that MeJA treatment severely affected the normal metabolism of ribosomal proteins of *G. lucidum*.

### 2.6. Transcriptional Expression Analysis of Selected DAPs as Revealed by qRT-PCR

To validate our proteomics data, qRT-PCR was used to determine the mRNA expression patterns of the 32 distinct DAPs from M15 and M24 mentioned above, in which 10 selected genes were from the energy metabolism pathway (Figure 4A), 10 selected genes were from transcriptional and translational regulation (Figure 4B), four selected genes were from the triterpenoid synthesis pathway (Figure 4C), and eight selected genes were from the oxidoreduction process (Figure 4D). The results show that the expression patterns of hexokinase-2 (HXK2) and glycerol 2-dehydrogenase (GCY1) were upregulated at M15 and downregulated at M24. The expression patterns of phosphoglucomutase 2 (PGM2), phosphoglycerate kinase 1 (PGK1), and phosphoenolpyruvate carboxy kinase 1 (PCK1) were upregulated both at M15 and M24 (Figure 4A). The triterpenoid synthesis genes, lanosterol 14-alpha demethylase ERG11 (GL28943-R1_1) and hydroxymethylglutaryl-CoA synthase ERG13 (GL24922-R1_1) were upregulated at both M15 and M24 (Figure 4C). The farnesyl pyrophosphate synthase ERG20 (GL22068-R1_1) and cytochrome P450 ERG5 (GL27042-R1_1) were upregulated at M15 and M24, respectively (Figure 4C). The expression patterns of the oxidoreduction process and transcriptional and translational regulation genes were also similar in their protein expression patterns (Figure 4B,D). However, the expression patterns of isocitrate lyase (ICL1) were downregulated at both M15 and M24, which was the opposite of their corresponding proteins (Figure 4A). The summarized primer data of 32 representative DAPs are shown in Appendix A. Overall, the expression levels of most of the genes are consistent with the protein levels.

### 2.7. Metabolic Changes in Response to MeJA

Compared to signal molecules and gene expression that can respond to MeJA in minutes, changes in metabolism generally take hours. Therefore, the untargeted GC–MS and LC–MS metabolomics analyses were used to further research the control and M24 samples to elucidate the modulation of metabolic processes in the *G. lucidum* response to MeJA. The principal component analysis (PCA) score plots show that the samples were apparently separated among the control group and the MeJA treatment group (Appendix A). PLS-DA and OPLS-DA analysis score plots also showed that the metabolic profiles were significantly changed between the control group and the MeJA treatment group (Appendix A). For the untargeted metabolomics analysis, we were able to assign structural identities to 224 metabolites (154 by GC–MS in Appendix A and 70 by LC-MS/MS in Appendix A) which displayed significant changes in levels (*p* < 0.05) between the different samples (M24/C24).

The KEGG analysis revealed that the 154 metabolites by GC–MS and 70 metabolites by LC–MS/MS were enriched in various cellular metabolism pathways. The most significant (*p* < 0.01) pathways of the 154 metabolites enriched by GC–MS were ABC transporters followed by processes such as the biosynthesis of plant secondary metabolites, the biosynthesis of amino acids, protein digestion and absorption, and purine metabolism (Figure 5A and Appendix A). The most significant (*p* < 0.01) pathways of the 70 metabolites enriched by LC–MS/MS were the biosynthesis of unsaturated fatty acids followed by fatty acid biosynthesis, purine metabolism, cutin/suberine/wax biosynthesis and biosynthesis of plant secondary metabolites (Figure 5B and Appendix A). The total of 224 metabolites enriched metabolite KEGG pathways was similar to the 202 DAPs of M24.

In detail, GC-MS detected a significant decrease in TCA cycle intermediates under MeJA treatment for 24 h (Appendix A), such as L-malic acid (0.823), fumaric acid (0.968), alpha-ketoglutaric acid (0.714), and pyruvic acid (0.521). These results also indicate that MeJA induction inhibited normal glucose metabolism, which was consistent with the results of proteomics. Moreover, some secondary metabolites such as alpha-hydroxycholesterol, zymosterol, ergosterol (Appendix A), and lepidiumsesterterpenol, 2-heptadecanone, tetrahydroxyergosta-7,22-dien-6-one, ergosta-4,6,8(14),22-tetraen-3-ol, 3,5,9-Trihydroxyergost-7-en-6-one (Appendix A) were detected and upregulated. These results further indicate that MeJA induction resulted in metabolic reprogramming which inhibited the normal glucose metabolism and energy supply and enhanced the secondary metabolism of cells compared with normal physiological levels in *G. lucidum*.

In addition, LC–MS detected a significant change in the content of 27 kinds of fatty acids under MeJA treatment for 24 h, in which 10 were upregulated and 17 were downregulated. Interestingly, most of the upregulated fatty acids were saturated, such as cerebronic acid, (Z)-13-octadecenoic acid, behenic acid and (R)-3-hydroxy-octadecanoic acid, and most of the downregulated fatty acids were unsaturated, such as 15,16-Epoxy-9,12-octadecadienoic acid, 6-hydroxy-9,12,14-octadecatrienoic acid, DG (14:1/22:6), and dihomo-linoleate (20:2n6) (Appendix A). In addition, the proteomic analysis showed that fatty aldehyde dehydrogenase (GL20474-R1_1) was downregulated by 0.386-fold after MeJA treatment for 24 h (Appendix A). These results indicate that MeJA treatment leads to a decrease in fatty acid unsaturation.

On the other hand, the metabonomic results reveal that the contents of substrate for protein synthesis and amino acids were upregulated under MeJA treatment for 24 h (Appendix A). The GC–MS results show that the contents of norleucine, norvaline, glutamine, *N*,*N*-dimethylarginine, cycloleucine, alanine, and lysine increased by more than 20 times, and cysteinyl glycine, glycine, serine, phenylalanine, and O-succinyl homoserine increased by more than two times. LC–MS results showed that the contents of isoleucyl valine increased 1.357-fold. These results further confirm that MeJA treatment severely inhibits the protein synthesis of *G. lucidum*.

## 3. Discussion

The biosynthesis of secondary metabolites of fungi is regulated by many environmental factors. MeJA is a type of plant hormone signaling molecule, which is ubiquitous in herbs and woody plants. During the development of mycelium and fruiting body, the wood-rotting fungus *G. lucidum* will almost inevitably encounter JA signal substances because it parasitizes on plant trunks in nature. Thus, in theory, *G. lucidum* should evolve the genetic basis of response to MeJA signal. In this study, we found that *G. lucidum* hyphae responded rapidly to MeJA signal. Many transcription factors had undergone changes in gene expression levels, and various changes had taken place at the protein and metabolic levels in M15 and M24. Our results will provide a reference for understanding the interaction mechanism between plant parasitic fungi and plants. In addition, terpenoids are the most abundant and structurally diverse group of plant and microorganism secondary metabolites [36]. MeJA has been successfully used as an effective elicitor to enhance the production of terpenoids in *G. lucidum* [25,26]. However, the underlying metabolic and regulatory mechanisms have not been determined to date. To this end, in this study, we integrated an analysis of proteomics and metabolomics to dissect the mechanism of global responses to MeJA in *G. lucidum*. We identified some differentially expressed proteins and metabolites involved in a variety of cellular processes, mainly including the oxidoreduction process, secondary metabolism, energy metabolism, transcriptional and translational regulation, and protein synthesis.

### 3.1. Oxidoreduction Process

The results of the proteomic analysis show that multiple functional proteins in the cell antioxidant and reductive system were upregulated, such as glutathione s-transferase L1, aryl-alcohol dehydrogenase, and cytochrome P450 oxidoreductase under MeJA treatment. The catalase and polyamine oxidase were downregulated under MeJA treatment. Previous studies have found that ROS signals produced by oxidoreductase imbalance were involved in GAs biosynthesis under MeJA, heat, and in vegetable oil treatments in *G. lucidum* [10,13,27]. In addition, cytochrome P450 catalysis participates in various important cellular/metabolic processes, including the biosynthesis of hormones, secondary metabolism, and fatty acid metabolism [37]. Therefore, our results further confirm the regulatory role of the redox system in GA biosynthesis in *G. lucidum*. Moreover, it has been proven that GAs are synthesized via the mevalonate pathway from glucose to lanosterol in *G. lucidum* [38]. Further biosynthetic steps from lanosterol to GAs include a series of reduction and/or oxidation of lanosterol by cytochrome P450 monooxygenase [4,39]. The results with upregulated cytochrome P450 oxidoreductase also further indicate that MeJA can promote the synthesis of triterpenoids.

On the other hand, glutathione dehydrogenase (GL29181-R1_1) and S-formylglutathione hydrolase (GL29422-R1_1) were downregulated under MeJA treatment for 24 h. In addition, the results of the metabolic analysis show that the content of glutathione was downregulated by GC–MS and LC–MS. The consistency between the downregulation of glutathione in the proteomic data and the decrease of the content of glutathione in the metabolomic data indicates that the cells are subjected to redox pressure under MeJA treatment. Previous studies have preliminarily revealed that the MeJA-induced ROS burst is partly dependent on NADPH oxidase [27]. Our results reveal that the reduction of the glutathione system may be another reason for the ROS burst by MeJA to regulate GA biosynthesis in *G. lucidum*.

### 3.2. Secondary Metabolism

After MeJA treatment, hydroxymethylglutaryl-CoA synthase (GL24922-R1_1), cytochrome P450 (GL27042-R1_1), and farnesyl pyrophosphate synthase (GL22068-R1_1) were upregulated by proteomic analysis, which indicates that terpene steroid secondary metabolism and the mevalonate pathway were significantly enhanced. In addition, GC-MS and LC-MS detected a significant increase in some secondary metabolites such as alpha-hydroxycholesterol, zymosterol, ergosterol, and 2-heptadecanone. The consistency between the upregulation of hydroxymethylglutaryl-CoA synthase, cytochrome P450, and farnesyl pyrophosphate synthase in proteomic data and the increase of the content of alpha-hydroxycholesterol, zymosterol, ergosterol, and 2-heptadecanone in metabolomic data further confirm the promoting effect of MeJA on the biosynthesis of GAs in *G. lucidum*.

In our previous studies, cDNA-Amplified Fragment Length Polymorphism (cDNA-AFLP) was used to identify genes that were differentially expressed in response to MeJA in *G. lucidum*, which found that cytochrome b2 and cytochrome P450 were upregulated [26]. In the research of Liang et al., the addition of phenobarbital, a cytochrome P450 inducer, could enhance the production of total and individual GAs in a two-stage cultivation involving a period of initial shake flask culture followed by static liquid culture of *G. lucidum* [8]. Chen et al. analyzed the members of the cytochrome family in *G. lucidum* genome and found that 78 CYP genes are co-expressed with lanosterol synthase, and 16 of these show a high similarity to fungal CYPs that specifically hydroxylate testosterone, suggesting their possible roles in triterpenoid biosynthesis [40]. In the studies of Wang et al., using *Saccharomyces cerevisiae* as a host, overexpression of a CYP450 gene *cyp5150l8* from *G. lucidum*, found that CYP5150L8 catalyzes a three-step biotransformation of lanosterol at C-26 to synthesize an antitumor GA, 3-hydroxy-lanosta-8, 24-dien-26 oic acid (HLDOA) [41]. In this study, we also found that the multiple cytochrome family proteins were significant increased, such as cytochrome b2 (by 1.941-fold in M15), cytochrome P450 61 (by 3.233-fold in M15), and cytochrome P450-DIT2 (by 2.42-fold in M24). All these results indicate that cytochrome family genes played an important role in the triterpenoid biosynthesis pathway, and the triterpenoids with different structures were related to the reaction catalyzed by cytochrome family members.

### 3.3. Energy Metabolism

The results of the proteomic analysis show that multiple functional proteins in glycolysis, sugar isogenesis, and fatty acid metabolism were upregulated under MeJA treatment for 15 min but downregulated under MeJA treatment for 24 h (Figure 6 and Appendix A). In addition, GC–MS detected a significant decrease in TCA cycle intermediates under MeJA treatment for 24 h, such as L-malic acid, fumaric acid, alpha-ketoglutaric acid, and pyruvic acid. These results indicate that MeJA induction inhibited normal glucose metabolism and suggest that the intracellular energy metabolism pathways were rearranged from basic metabolism to secondary metabolism in the MeJA induction proceeding (Figure 6 and Appendix A).

On the other hand, LC–MS detected a significant change in the content of 27 kinds of fatty acids under MeJA treatment for 24 h, in which 10 were upregulated and 17 were downregulated. Interestingly, most of the upregulated fatty acids were saturated, such as cerebronic acid, (Z)-13-octadecenoic acid, behenic acid and (R)-3-hydroxy-octadecanoic acid, and most of the downregulated fatty acids were unsaturated, such as 15,16-Epoxy-9,12-octadecadienoic acid, 6-hydroxy-9,12,14-octadecatrienoic acid, DG (14:1/22:6), and dihomo-linoleate (20:2n6). In addition, the proteomic analysis showed that fatty aldehyde dehydrogenase (GL20474-R1_1) was downregulated in M24. The consistency between the downregulation of fatty aldehyde dehydrogenase in the proteomic data and the decrease of the unsaturated fatty acids content in the metabolomic data indicates that MeJA treatment leads to a decrease in fatty acid unsaturation. It is a universal law of biological cells, of which *G. lucidum* is no exception, that an increase of membrane unsaturation degree due to an increase of unsaturated fatty acids leads to enhanced membrane fluidity [42]. In addition, membrane fluidity is involved in the regulation of GA biosynthesis [43]. Moreover, a similar result also appears in plants that MeJA treatments could enhance chilling tolerance in fruits and vegetables under postharvest by increasing membrane fluidity [44]. Therefore, the regulatory mechanism of MeJA-induced GA biosynthesis may involve membrane fluidity and fatty acid unsaturation, which warrants further investigation.

Moreover, GC–MS results show that the cyclic AMP content increased by 224,866.4-fold, and iTRAQ results showed that the cAMP-dependent protein kinase regulatory subunit (GL20414-R1_1) was upregulated by 1.697-fold under MeJA treatment for 24 h. These results show that the upregulation of cAMP-dependent protein kinase regulatory subunit in proteomic data is consistent with the increase of cyclic AMP content in metabolomic data. In *Arabidopsis thaliana*, treatment with MeJA for 3 min resulted in an increase in intracellular cAMP content to peak value and then activated the cell membrane cyclic nucleotide gated ion channel 2 (AtCNGC2), resulting in extracellular Ca^2+^ influx [45]. Previous studies have found that intracellular calcium signals regulate the biosynthesis of GAs under heat stress [11,12]. Therefore, a potential mechanism was that cAMP might activate the intracellular calcium signal and then regulate the biosynthesis of GAs under MeJA.

### 3.4. Protein Synthesis

Ribosomes are the sites of protein synthesis. The proteomic data show that the contents of multiple ribosome proteins and the chaperones (heat shock protein) were downregulated under M15 and M24. In addition, transcriptional and translation-related regulatory factors were downregulated under MeJA treatment. Moreover, multiple functional proteins in protein synthesis, folding and transport were also downregulated under M15 and M24. These results indicate that MeJA treatment severely inhibits the protein synthesis of *G. lucidum.* The metabonomic results reveal that the contents of substrate for protein synthesis and amino acids were upregulated under MeJA treatment for 24 h. The GC–MS results show that the contents of norleucine, norvaline, glutamine, *N*,*N*-dimethylarginine, cycloleucine, alanine, and lysine increased by more than 20 times, and cysteinyl glycine, glycine, serine, phenylalanine, and O-succinyl homoserine increased by more than twice. The LC–MS results show that the contents of isoleucyl valine increased by 1.357-fold. The correlation between the downregulation of ribosome proteins, the chaperones, multiple functional proteins in protein synthesis, folding and transport in proteomic data, and the increase of the contents of amino acids, norleucine, norvaline, glutamine, *N*,*N*-dimethylarginine, cycloleucine, alanine, and lysine in metabolomic data further confirm that MeJA treatment severely inhibits the protein synthesis of *G. lucidum*. It has been reported that promoting apoptosis, interfering with mitochondrial homeostasis or inhibiting the division cycle is the molecular mechanism by which MeJA kills cancer cells or plant-resistant pathogens [46]. We also found that some cell division control protein decreased, which suggests that the cell cycle might be delayed by the MeJA. However, almost no research has found that MeJA could resist pathogens or kill cancer cells by inhibiting protein synthesis. Our results also provide a new possible explanation that *G. lucidum* is easy to grow on dead trees, rather than living trees, because MeJA severely inhibits protein synthesis.

In conclusion, we performed integrated proteomics and metabolomics analyses to examine the different molecular mechanisms in the *G. lucidum* response to MeJA in this study. Our observations successfully identified 209 DAPs in M15 and 202 DAPs in M24, as well as 154 metabolites by GC–MS and 70 metabolites by LC–MS involved in several metabolic pathways. The results further confirm the promoting effect of MeJA on the biosynthesis of GAs in *G. lucidum.* Our research analyzed and discussed some differentially expressed proteins and metabolites involved in the oxidoreduction process, secondary metabolism, energy metabolism, transcriptional and translational regulation, and protein synthesis in depth. Our results reveal that MeJA inhibited the normal glucose metabolism, energy supply, and protein synthesis but promoted secondary metabolites, including GAs in *G. lucidum*. In addition, our results indicate that the reduction of the glutathione system is another reason for the ROS burst by MeJA. Moreover, MeJA regulates GA biosynthesis may through cAMP activation of the intracellular calcium signal or fatty acid unsaturation and membrane fluidity, which warrant further research. In conclusion, our proteomics and metabolomics data will provide a valuable resource for further investigation of the molecular mechanisms of MeJA signal response and GA biosynthesis in *G. lucidum* and other related species.

## 4. Materials and Methods

### 4.1. Fermentation Conditions and Methyl Jasmonate Elicitation of *G. lucidum* (Fungal Culture and Sample Preparation)

The *G. lucidum* strain was obtained from the Agricultural Culture Collection of China, ACCC53264). *G. lucidum* was grown at 28 °C in potato dextrose agar (PDA) solution medium with 50 μM methyl jasmonate (MeJA, Sigma-Aldrich, St. Louis, MO, USA) for 15 min and 24 h (M15 and M24). The PDA medium without MeJA instead of equal volumes of ethanol served as the control (C15 and C24). MeJA induction was dissolved in ethanol and sterilized with a 0.2 μm Supor Membrane Acrodisc Syringe Filter (PALL, Port Washington, NY, USA) before addition to the medium. The fermentation conditions of *G. lucidum* were maintained as described [26].

### 4.2. Protein Extraction

*G. lucidum* mycelia (approximately 1.0 g) were homogenized in a liquid nitrogen-chilled cryogenic mill and the powders were precipitated in a 10% (*w*/*v*) trichloroacetic acid/acetone solution, containing 65 mM dithiothreitol (DTT) at −20 °C for 1 h. Then homogenates were centrifuged (11,000× *g*, 30 min, 4 °C) and rinsed with ice-cold acetone for twice. Then, protein pellets were vacuum dried and ready for use. For each sample, the pellets were dissolved in lysis solution containing 7 M urea, 2 M thiourea, 4% *w*/*v* CHAPS, 1% *w*/*v* DTT, 1 mM PMSF, and 0.5% *v*/*v* biolytic (Bio-Rad, Hercules, CA, USA). Impurities were removed by centrifugation and subsequent supernatant was collected. The protein concentration was quantified using the Bradford Assay Kit (Applied Biosystems, Waltham, MA, USA). Finally, the quality of proteins was evaluated by 12% SDS-PAGE.

### 4.3. iTRAQ Labelling and Strong Cation Exchange Fractionation

In this study, protein samples were reduced, alkylated, and trypsin-digested. Following the iTRAQ, labeling was performed using an iTRAQ Reagents 8-plex Kit (AB Sciex, Inc., Foster City, CA, USA) according to the manufacturer’s instructions. The specific content is as follows: the control for 15 min replicates and 24 h replicates were labelled with 115, 116 and 113, 114, respectively. Their treatment groups were labelled with 117, 118 and 119, 121, respectively. After labelling, the peptide samples were mixed and lyophilized. Then, the mixed peptide samples were dissolved in strong cation exchange (SCX) buffer A (2% (*v*/*v*) acetonitrile, pH 10.0). The peptides were fractionated on an Agilent HPLC system 1100 using a polysulfoethyl column/Narrow-Bore column (2.1 × 150 mm, 5 μm, 300 Å; Poly LC, Columbia, MD). Peptides were eluted at a flow rate of 300 μL/min with a linear gradient of 0%–20% solvent B (98% (*v*/*v*) acetonitrile, pH 10.0) for 50 min followed by ramping up to 100% solvent B for 5 min and holding for 10 min. The absorbance at 214 nm was monitored, and a total of 12 fractions were collected and vacuum dried.

### 4.4. Database Search and Quantitative Proteomic Analysis

The data were analyzed by Protein Pilot Software v. 5.0 (AB SCIEX, Framingham, MA, USA). Protein identification was performed using the transcriptome database of *G. lucidum*. Fixed modifications of methyl methane thiosulfate-labelled cysteine, fixed iTRAQ modification of free amine in the N terminus and lysine, and variable iTRAQ modifications of tyrosine were considered. Parameters such as trypsin digestion, precursor mass accuracy, and fragment ion mass accuracy are built-in settings of the software. The raw peptide identification results from the paragon algorithm were further processed by the Pro Group TM algorithm. The Pro Group algorithm uses the peptide identification results to determine the minimal set of confident proteins. For each protein identification, two types of scores are reported, i.e., unused Prot Score and total Prot Score. The total Prot Score is a measurement of all the peptide evidence for a protein and is analogous to protein scores reported by other protein identification software. The unused Prot Score is a measurement of all the peptide evidence for a protein that is not better explained by a higher-ranking protein. The unused Prot Score prevents the reuse of the same peptide evidence to support the detection of more than one protein. Thus, this score is the real indicator of protein confidence. The software calculates a percentage of confidence that reflects the probability that the hit is a false positive so that at the 99% confidence level, there is a false positive identification rate of 1%. Low confidence peptides do not identify a protein by themselves but support the identification of the protein. For proteins with only one significant contributing peptide, the MS/MS spectrum of the peptide was manually inspected and confirmed. The false discovery level was estimated by performing the search against a concatenated database containing both forward and reversed sequences. For protein relative quantification using iTRAQ, only MS/MS spectra unique to a particular protein and for which the sum of the signal-to-noise ratio for all of the peak pairs was greater than nine were used for quantification (software default settings, Applied Biosystems). The mean, S.D., and *p* values to estimate the statistical significance of the protein changes were calculated by Pro Group. For the identification of expression differences, each experimental run was initially considered separately. To be identified as being differentially expressed, a protein had to be quantified with at least three spectra (allowing the generation of a *p* value), a *p* value ≤ 0.05, and a ratio fold change of at least 2 in more than two independent experiments (i.e., at least six peptides).

### 4.5. Bioinformatic Analysis of DAPs

In the present study, a multi-omics data analysis tool, OmicsBean (http://www.omicsbean.cn), which integrated Gene Ontology (GO) enrichment, Kyoto Encyclopedia of Genes and Genomes (KEGG) pathway analysis, was employed to analyze the obtained DAPs. A Venn Diagram online tool (http://bioinformatics.psb.ugent.be/webtools/Venn/) was used to locate the co-existing DAPs among M15 and M24.

### 4.6. RNA Extraction Procedure and cDNA Synthesis

For each sample, about 0.5 g of mycelia was collected by filtration from the culture media, dehydrated in liquid nitrogen and stored at −80 °C for subsequent experiments. Total RNA was extracted using an RNA Isolation Kit (Takara, Dalian, China) and treated with DNase I (Takara, China) according to the manufacturer’s instructions. Double-stranded cDNA was synthesized from 2.5 mg total RNA using an M-MLV RTase cDNA Synthesis Kit (TaKaRa) and an oligo-dT primer (TaKaRa).

### 4.7. Real-Time RT-PCR Analysis

Real-time PCR was performed on pools of RNA derived from two independent biological experiments. All the samples were examined in triplicate. The samples were prepared as described above for iTRAQ. Then, 5 mL of 1:10 diluted cDNA samples was used as the qRT-PCR template with 0.5 mM gene-specific primers and 10 mL SYBR Premix Ex Taq II (Takara, China) in a total volume of 20 mL. Experiments were performed in a Real plex 2 Systems (Eppendorf, Germany) with the following thermal cycling profile: 95 °C for 10 min, followed by 40 cycles of 95 °C for 30 s, 55 °C for 30 s, and 72 °C for 30 s. Each real-time assay was tested in a dissociation protocol to ensure that each amplicon was a single product. The relative quantification of gene expression was performed using the housekeeping gene PP2A [47]. Specific primer pairs were designed for the target genes chosen for validation using the Primer 5 software (Appendix A). Post-reverse transcription quantitative PCR (qRT-PCR) calculations analyzing the relative gene expression levels were performed according to the 2^−∆∆*C*t^ method.

### 4.8. GC-MS Data Preprocessing, Statistical Analysis, and Identification of Differential Metabolites

The derivatized samples were analyzed on an Agilent 7890B gas chromatography system coupled to an Agilent 5977A MSD system (Agilent, Palo Alto, CA, USA). A DB-5MS fused-silica capillary column (30 m × 0.25 mm × 0.25 μm, Agilent J & W Scientific, Folsom, CA, USA) was utilized to separate the derivatives. Helium (>99.999%) was used as the carrier gas at a constant flow rate of 1 mL/min through the column. The injector temperature was maintained at 260 °C. The injection volume was 1 μL in split mode (split ratio is 2:1). The initial oven temperature was 90 °C, ramped to 180 °C at a rate of 10 °C/min, to 240 °C at a rate of 5 °C/min, to 290 °C at a rate of 25 °C/min, and finally held at 290 °C for 11 min. The temperature of the MS quadrupole, and ion source (electron impact) were set to 150 and 230 °C, respectively. The collision energy was 70 eV. Mass data were acquired in full-scan mode (*m*/*z* 50–450), and the solvent delay time was set to 5 min.

The QCs were injected at regular intervals (every six samples) throughout the analytical run to provide a set of data from which repeatability can be assessed. The acquired MS data from GC−MS were analyzed by ChromaTOF software (v 4.34, LECO, St Joseph, MI). Metabolites were measured by the NIST and Fiehn database, which is linked to the ChromaTOF software. The resulting data were normalized to the total peak area of each sample, multiplied by 10,000, transformed by log2 in Excel 2007 (Microsoft, Redmond, WA, USA) and imported into a SIMCA (version 14.0, Umetrics, Umeå, Sweden), where principal component analysis (PCA), partial least-squares discriminant analysis (PLS-DA), and orthogonal partial least-squares discriminant analysis (OPLS-DA) were performed. The Hotelling’s T2 region, shown as an ellipse in score plots of the models, defines the 95% confidence interval of the modelled variation. The quality of the models is described by the R2X or R2Y and Q2 values. R2X or R2Y is defined as the proportion of variance in the data explained by the models and indicates goodness of fit. Q2 is defined as the proportion of variance in the data predicted by the model and indicates predictability, calculated by a cross-validation procedure. A default seven-round cross-validation in SIMCA was performed to determine the optimal number of principal components and to avoid model overfitting. The OPLS-DA models were also validated by a permutation analysis (200 times).

The differential metabolites were selected on the basis of the combination of a statistically significant threshold of variable influence on projection (VIP) values obtained from the OPLS-DA model and *p* values from a two-tailed Student’s *t*-test on the normalized peak areas, where metabolites with VIP values larger than 1.0 and *p* values less than 0.05 were included, respectively.

### 4.9. LC-MS/MS Data Pre-Processing, Statistical Analysis, and Identification of Differential Metabolites

An ACQUITY UHPLC system Ultimate 3000 (Thermo Fisher Scientific, Waltham, MA, USA) coupled with LTQ Orbitrap MS (Thermo Fisher Scientific, Waltham, MA, USA) was used to analyze the metabolic profiling in both ESI positive and ESI negative ion modes. In positive ion mode, the separation of metabolites was conducted on a 2.1 × 100 mm ACQUITYTM 1.7 μm BEH C8 column, and the mobile phase contained water with 0.1% formic acid (A) and acetonitrile (B). The linear elution gradient programme was used as follows: 5% B was kept for 1.0 min, then linearly increased to 100% B at 24 min, held for 4 min, 100%–5% B from 28 to 28.1 min, and held at 5% from 28.1 to 30 min. Each run time was 30 min. In negative ion mode, the metabolite separation was performed on a 2.1 × 100 mm ACQUITYTM 1.8 μm HSS T3 column, and the mobile phase contained 6.5 mM ammonium bicarbonate water solution (C) and 6.5 mM ammonium bicarbonate in 95% methanol and water (D). The linear elution gradient programme was 5% D kept for 1.0 min, then linearly increased to 100% D at 18 min, held for 4 min, 100%–5% B from 22 to 22.1 min, and held at 5% from 22.1 to 25 min. Each run time was 25 min. The flow rate was 0.35 mL/min, and the column temperature was 50 °C. The injection volume was 5 μL. Mass spectrometry detections were set as the following: capillary temperature 350 and 360 °C, spray voltage 3.5 and 3.0 kV for positive ion mode and negative ion mode, respectively. The mass scan range was *m*/*z* 50 to 1000. The resolution of the MS was set to 30,000. The QCs were injected at regular intervals (every 10 samples) throughout the analytical run to provide a set of data from which repeatability can be assessed.

The acquired MS data from UHPLC-LTQ Orbitrap were analyzed by the software XCMS, which produced a matrix of features with the associated retention time, accurate mass and chromatographic. The variables that presented in least 80% of either group were extracted. The variables with <30% relative standard deviation (RSD) in QC samples were then retained for further multivariate data analysis because they were considered to be stable enough for prolonged UHPLC-LTQ analysis. The internal peaks were removed from the data set. The resulting data were normalized to the total peak area of each sample in Excel 2007 (Microsoft, Redmond, WA, USA).

Data were imported into a SIMCA (version 14.0, Umetrics, Umeå, Sweden), where PCA, PLS-DA, and OPLS-DA were performed. The Hotelling’s T2 region, shown as an ellipse in the score plots of the models, defines the 95% confidence interval of the modelled variation. The quality of the models is described by the R2X or R2Y and Q2 values. R2X or R2Y is defined as the proportion of variance in the data explained by the models and indicates goodness of fit. Q2 is defined as the proportion of variance in the data predicted by the model and indicates predictability, calculated by a cross-validation procedure. A default seven-round cross-validation in SIMCA was performed throughout to determine the optimal number of principal components and to avoid model overfitting. The OPLS-DA models were also validated by a permutation analysis (200 times).

The differential metabolites were selected on the basis of the combination of a statistically significant threshold of variable influence on projection (VIP) values obtained from the OPLS-DA model and *p* values from a two-tailed Student’s *t*-test on the normalized peak areas, where metabolites had VIP values larger than 1.5 and *p* values less than 0.05, respectively.

## Figures and Tables

**Figure 1 ijms-20-06116-f001:**
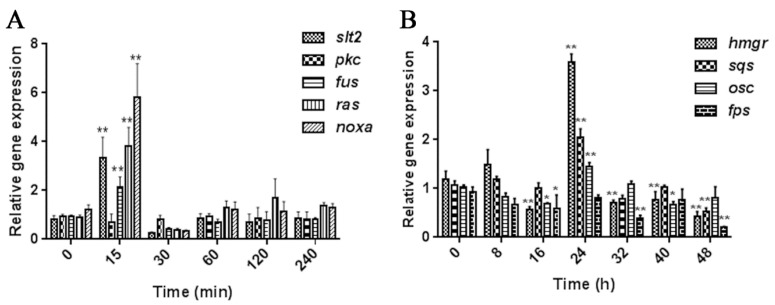
Expression levels of selected signal transduction and ganoderic acids (GA) biosynthetic pathway genes under methyl jasmonate (MeJA) induction. (**A**) Relative expression levels of signal transduction genes *slt2*, *pkc*, *fus*, *ras*, and *noxa* under 50 μM MeJA treatment for 0–240 min. (**B**) Relative expression levels of key genes of the GAs biosynthetic pathway *hmgr*, *sqs*, *osc*, and *fps* under 50 μM MeJA treatment for 0–48 h. Each significance test of the difference (*t*-test) was between the control and the MeJA treatment in each group. The values are the mean ± SD of three independent experiments. The asterisks indicate significant differences compared to the strains that were without MeJA treatment (Student’s *t*-test: * *p* < 0.05, ** *p* < 0.01).

**Figure 2 ijms-20-06116-f002:**
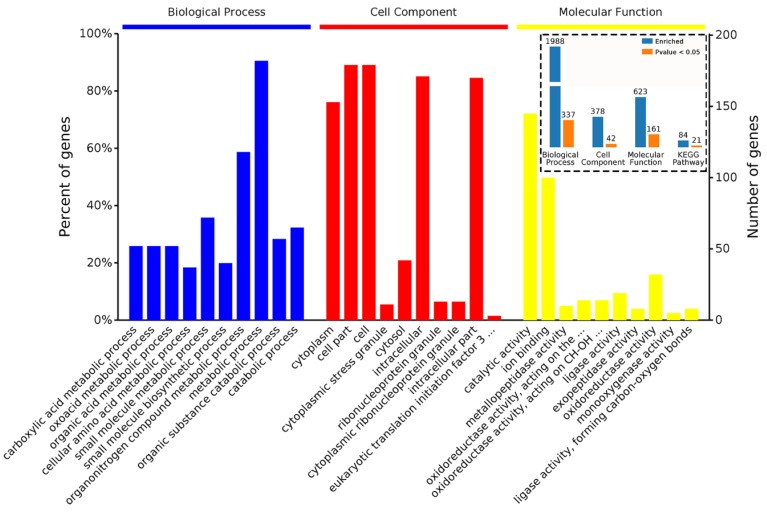
Bioinformatics analysis of 202 identified differential abundance proteins (DAPs) in MeJA treatment for 24 h (M24). BP, CC, MF, KEGG are four categories of functional analysis, that stand for biological process, cell component, molecule functions and KEGG pathway, respectively. Counts for each category represent the total associated terms in the database with the query protein list. Terms with a *p*-value < 0.05 are statistically significant. The 10 most significantly enriched terms in the level 4 gene ontology hierarchy, information on the percentage, and number of involved proteins in a term are shown on the left and right y-axes, respectively.

**Figure 3 ijms-20-06116-f003:**
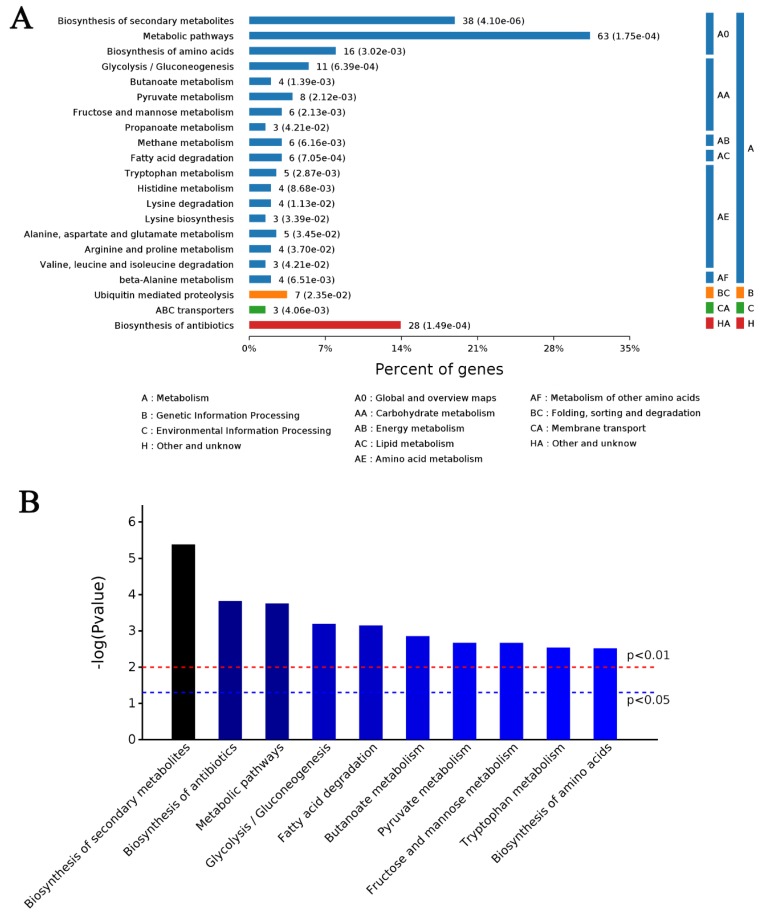
KEGG analysis of 202 identified DAPs in M24. (**A**) Enriched KEGG pathways are clustered into the metabolism subcategories, the number of involved proteins in a specific pathway and the corresponding *p*-value are shown on the right side of column. (**B**) The most significant 10 KEGG pathways.

**Figure 4 ijms-20-06116-f004:**
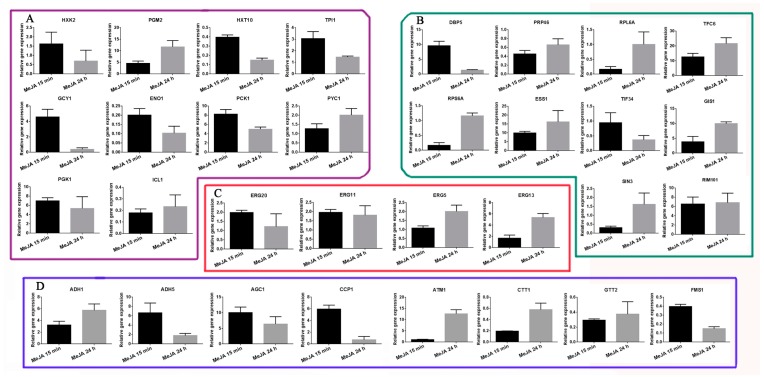
Analysis of the relative gene expression levels of some identified DAPs. qRT-PCR analyses of 10 selected genes in the energy metabolism pathway (**A**) (framed in purple), 10 selected genes in transcriptional and translational regulation (**B**) (framed in green), four selected genes in the triterpenoid synthesis pathway (**C**) (framed in red), and eight selected genes in the oxidoreduction process (**D**) (framed in blue).

**Figure 5 ijms-20-06116-f005:**
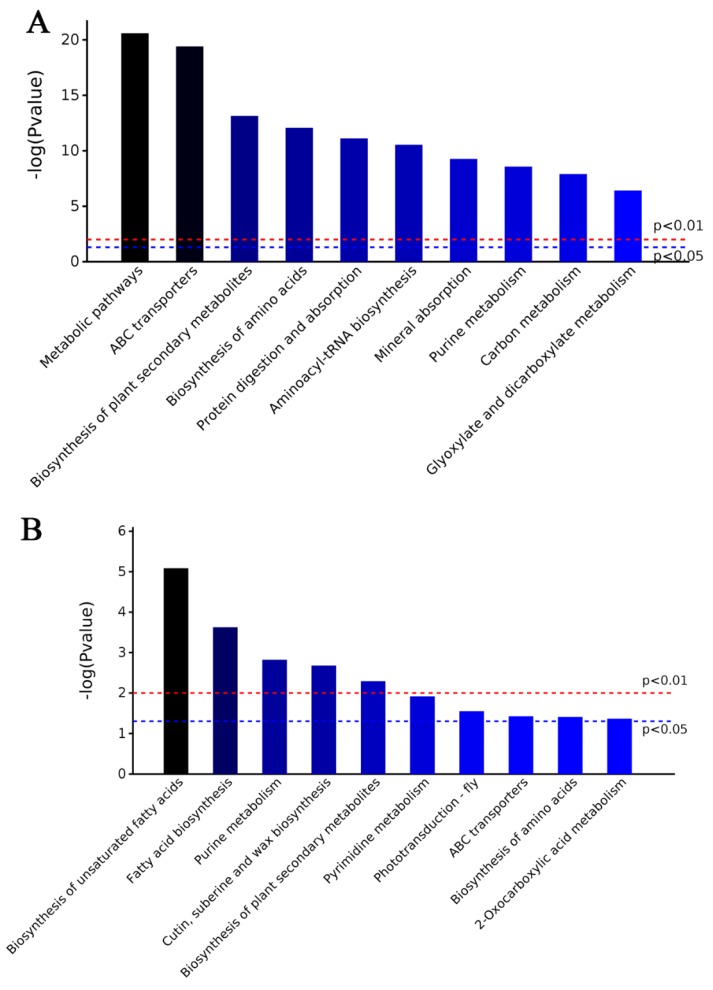
KEGG analysis of identified metabolites in M24. (**A**) The 10 most significant pathways enriched by KEGG of 154 metabolites by GC–MS. (**B**) The 10 most significant pathways enriched by KEGG of 70 metabolites by LC–MS.

**Figure 6 ijms-20-06116-f006:**
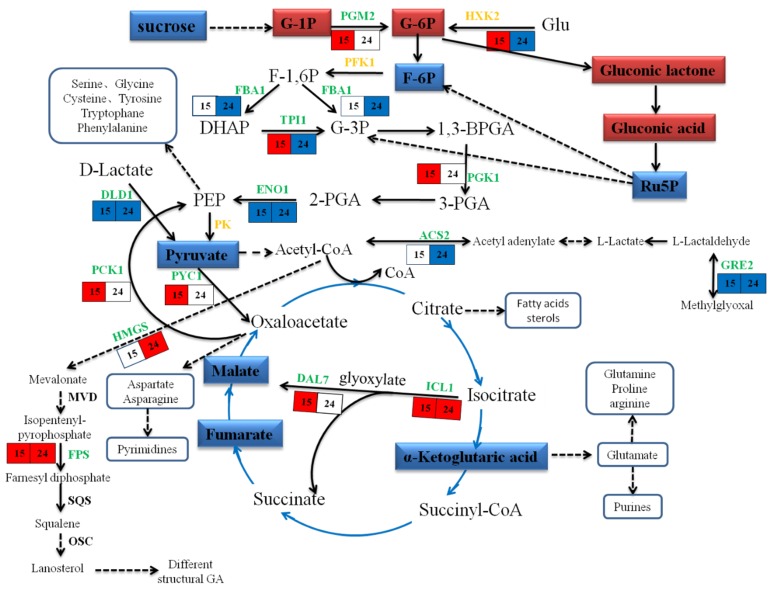
Integrated analysis GA biosynthesis and energy metabolism based on proteomic and metabolomic data. The stereoscopic frame shape indicates metabolites, in which red indicates upregulation, and blue indicates downregulation. Plane frames indicate proteins. Numbers 15 and 24 indicate MeJA treatment for 15 min and 24 h, respectively, in which red indicates upregulation, blue indicates downregulation, and white indicates not detected. The green font indicates a detected functional protein, and the golden indicates a rate-limiting enzyme. The solid line represents the one-step reaction and the dashed line represents the multi-step reaction.

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
