# Peer review of "Integrated Proteomics and Metabolomics Analysis Provides Insights into Ganoderic Acid Biosynthesis in Response to Methyl Jasmonate in Ganoderma Lucidum"

_ijms, 2019, doi:10.3390/ijms20246116_

Round 1

Reviewer 1 Report

The article by Jiang et al. is very interesting, well written and within the scope of the Journal.

While it could be published as it is after minor editorial changes (i.e. the formatting of the introduction, capital letters in some references across the manuscript and minor language polishing etc.), the core of the discussion, which is the comparison between proteomic and metabolomic data, could be improved with statistical analysis that could correlate the information of the two datasets.

Author Response

Dear Editor,

Thank you very much for your kind letter and the reviewers’ comments concerning our manuscript entitled “Integrated proteomics and metabolomics analysis provide insights into ganoderic acid biosynthesis in response to methyl jasmonate in Ganoderma lucidum” (ID: IJMS-642095). The comments were valuable and very helpful for revising and improving our paper as well as for guiding our research. We have thoroughly considered the comments and substantially revised our manuscript. Some new data were added in the revised manuscript, and the major revisions were marked in red colour in revised MS (In addition, the MS has undergone English language editing by MDPI which using the "Track Changes" function in the revised MS). The point-by-point replies to the reviewers’ comments are listed below:

Reviewers' Comments & Our Responses:

Review 1: The article by Jiang et al. is very interesting, well written and within the scope of the Journal. While it could be published as it is after minor editorial changes (i.e. the formatting of the introduction, capital letters in some references across the manuscript and minor language polishing etc.), the core of the discussion, which is the comparison between proteomic and metabolomic data, could be improved with statistical analysis that could correlate the information of the two datasets.

Response: Thanks for your positive comments and suggestions. According to the Reviewer's suggestion, we modified the formatting of the introduction, checked the capital letters in some references across the manuscript and we polished up the language too. Meanwhile, we have supplemented and improved the discussion part in Revised MS which marked in red colour in the new line 443-444, 455-457, 500-501, 512-514 and 532-535.

Review 2: This paper describes an integrated proteomics and metabolomics analysis of methyl jasmonate treatment and response by Ganoderma lucidum. The analyses were performed well and the resulting data described well.

Response: Thanks for your positive comments.

However, I recommend that the authors:

1. Show a biosynthetic pathway for ganoderic acid that clearly shows the relationship between GA biosynthesis and the energy metabolism pathways that were found to respond to MeJA treatment.

Response: Thanks for your suggestion. We have modified the previous Fig. 6 into the new Fig. 6 to shows the relationship between GA biosynthesis and the energy metabolism pathways. In addition, the corresponding figure legend has been modified in the Revised MS which marked in red colour in the new line 486.

Provide the rationale for the selection of the 5 signal transduction genes used to monitor the early responses to MeJA treatment.

Response: Thanks for your suggestion. We have provided the rationale for the selection of the 5 signal transduction genes in Revised MS which marked in red colour in the new line 88-89 and 97-99. Previous studies have found that slt2 is involved in the regulation of mycelium growth, fruiting body development, cell wall integrity and oxidative stress, as well as in the synthesis of ganoderma acid [1]. fus is a member of the serine/threonine-specific kinase mitogen-activated protein kinase family, which plays an important role in the early response and regulation of external signals [2]. As an important component of DAG/PKC signal transduction pathway, pkc plays an important role in intracellular signal regulation network [3]. ras is involved in the early response to extracellular signalling and multi-stress tolerance [4]. noxa is one of the subunits of membrane-bound fungal NADPH oxidases (Nox) that is involved in the regulation of ROS synthesis and defensive reactions [5]. Therefore, we selected these genes to monitor the early events of the G. lucidum response to MeJA.

Zhang, G., et al., The mitogen-activated protein kinase GlSlt2 regulates fungal growth, fruiting body development, cell wall integrity, oxidative stress and ganoderic acid biosynthesis in Ganoderma lucidum. Fungal Genet Biol, 2017. 104: p. 6-15. Maeder, C.I., et al., Spatial regulation of Fus3 MAP kinase activity through a reaction-diffusion mechanism in yeast pheromone signalling. Nat Cell Biol, 2007. 9(11): p. 1319-26. Heinisch, J.J. and R. Rodicio, Protein kinase C in fungi-more than just cell wall integrity. FEMS Microbiol Rev, 2018. 42(1). Guan, Y., et al., A novel Ras GTPase (Ras3) regulates conidiation, multi-stress tolerance and virulence by acting upstream of Hog1 signaling pathway in Beauveria bassiana. Fungal Genet Biol, 2015. 82: p. 85-94. Park, J.C., Y. Kim, and H.S. Lee, Involvement of the NADH oxidase-encoding noxA gene in oxidative stress responses in Corynebacterium glutamicum. Appl Microbiol Biotechnol, 2015. 99(3): p. 1363-74. Provide explanation as to why different time scales were used to analyse expression of signal transduction genes (min) vs GA biosynthetic pathway genes (h)

Response: Thanks for your suggestion. We have provided explanation in Revised MS which marked in red colour in the new line 97-99 and 109-111.

Move Tables 1, 2, 3, 4 and 5 to supplemental materials

Response: Thanks for your suggestion. We moved the Tables 1, 2, 3, 4, 5 and 6 to supplemental materials as Table S2, S3, S4, S5, S6 and S7. In addition, the corresponding result has been modified in the Revised MS which marked in red colour.

Explain whether they measured MeJA levels throughout the 24 h period. Was MeJA metabolized? If so, how long were the cells exposed to MeJA (if metabolized was this just a very short exposure?)

Response: Thanks for your question. The levels of MeJA was not measured in this manuscript. However, in our previous work, the MeJA levels have been measured in G. lucidum (Shi et al., 2015). In our previous work, the MeJA content in G. lucidum mycelia without MeJA treatment was 1.24 ± 0.16 ng/g fw, which was significantly increased to 22.21 ± 0.92 ng/g fw after 50μM MeJA treated 7-days (Shi et al., 2015). Therefore, when the cells exposed to MeJA, the endogenous MeJA has been up-regulated in G. lucidum mycelia even after MeJA treated seven days.

Shi, L., et al., The regulation of methyl jasmonate on hyphal branching and GA biosynthesis in Ganoderma lucidum partly via ROS generated by NADPH oxidase. Fungal Genet Biol, 2015. 81: p. 201-11.

Condense Figure 2 into 1 figure or eliminate Fig. 2A

Response: Thanks for your suggestion. We have modified the previous Fig. 2 into the new Fig. 2. In addition, the corresponding figure legend has been modified in the Revised MS which marked in red colour in the new line 216-222.

We hope, with these modifications, the quality of our manuscript would meet the publication standard of the Journal. We appreciate very much for your time in editing our manuscript, and the detailed and useful comments and suggestions from you and reviewers. I am looking forward to hearing from your suggestion about our revised manuscript.

Sincerely yours,

Ming-Wen Zhao

E-mail: mwzhao@njau.edu.cn

Tel: +0086-25-84395602

Fax: +0086-25-84395602

Reviewer 2 Report

This paper describes an integrated proteomics and metabolomics analysis of methyl jasmonate treatment and response by Ganoderma lucidum. The analyses were performed well and the resulting data described well. However, I recommend that the authors:

Show a biosynthetic pathway for ganoderic acid that clearly shows the relationship between GA biosynthesis and the energy metabolism pathways that were found to respond to MeJA treatment Provide the rationale for the selection of the 5 signal transduction genes used to monitor the early responses to MeJA treatment. Provide explanation as to why different time scales were used to analyse expression of signal transduction genes (min) vs GA biosynthetic pathway genes (h) Move Tables 1, 2, 3, 4 and 5 to supplemental materials Explain whether they measured MeJA levels throughout the 24 h period. Was MeJA metabolized? If so, how long were the cells exposed to MeJA (if metabolized was this just a very short exposure?) Condense Figure 2 into 1 figure or eliminate Fig. 2A

Author Response

(The authors gave the same response as above.)
